# GlanceNets: Interpretable, Leak-proof Concept-based Models

**Emanuele Marconato**
Department of Computer Science
University of Pisa & University of Trento
Pisa, Italy
emanuele.marconato@unitn.it

**Andrea Passerini**
Department of Computer Science
University of Trento
Trento, Italy
andrea.passerini@unitn.it

**Stefano Teso**
Department of Computer Science
University of Trento
Trento, Italy
stefano.teso@unitn.it

## Abstract

There is growing interest in concept-based models (CBMs) that combine high-performance and interpretability by acquiring and reasoning with a vocabulary of high-level concepts. A key requirement is that the concepts be interpretable. Existing CBMs tackle this desideratum using a variety of heuristics based on unclear notions of interpretability, and fail to acquire concepts with the intended semantics. We address this by providing a clear definition of interpretability in terms of alignment between the model's representation and an underlying data generation process, and introduce GlanceNets, a new CBM that exploits techniques from causal disentangled representation learning and open-set recognition to achieve alignment, thus improving the interpretability of the learned concepts. We show that GlanceNets, paired with concept-level supervision, achieve better alignment than state-of-the-art approaches while preventing spurious information from unintendedly leaking into the learned concepts.

## 1 Introduction

Concept-based models (CBMs) are an increasingly popular family of classifiers that combine the transparency of white-box models with the flexibility and accuracy of regular neural nets [1–5]. At their core, all CBMs acquire a vocabulary of concepts capturing high-level, task-relevant properties of the data, and use it to compute predictions and produce faithful explanations of their decisions [6].

The central issue in CBMs is how to ensure that the concepts are *semantically meaningful* and *interpretable* for (sufficiently expert and motivated) human stakeholders. Current approaches struggle with this. One reason is that the notion of interpretability is notoriously challenging to pin down, and therefore existing CBMs rely on different heuristics—such as encouraging the concepts to be sparse [1], orthonormal to each other [5], or match the contents of concrete examples [3]—with unclear properties and incompatible goals. A second, equally important issue is *concept leakage*, whereby the learned concepts end up encoding spurious information about unrelated aspects of the data, making it hard to assign them clear semantics [7]. Notably, even concept-level supervision is insufficient to prevent leakage [8], cf. Fig. 3.

36th Conference on Neural Information Processing Systems (NeurIPS 2022).

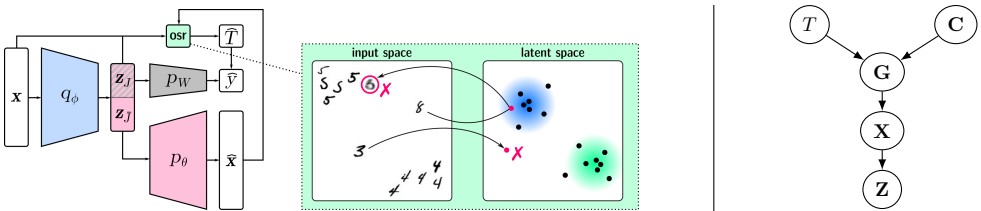

Figure 1: **Left**: Architecture of GlanceNets showing the encoder $q_\phi$, decoder $p_\theta$, classifier $p_W$, and open-set recognition step. **Center**: GlanceNets prevent leakage by identifying and rejecting open-set inputs using a combined strategy, shown here for a model trained on digits "4" and "5" only: the "3" is rejected as its embedding falls far away from classes prototypes (colored blobs), while the "8" is rejected as its reconstruction loss is too large. **Right**: The data generation process.

Prompted by these observations, we define interpretability in terms of *alignment*: learned concepts are interpretable if they can be mapped to a (partially) interpretable data generation process using a transformation that preserves semantics. This is sufficient to unveil limitations in existing strategies, build an explicit link between interpretability and disentangled representations, and provide a clear and actionable perspective on concept leakage. Building on our analysis, we also introduce GlanceNets (aliGned LeAk-proof coNCEptual Networks), a novel class of CBMs that combine techniques from *disentangled representation learning* [9] and *open-set recognition* [10] to actively pursue alignment – and guarantee it under suitable assumptions – and avoid concept leakage.

## 2    Concept-based Models

Concept-based models (CBMs) comprise two key elements: (i) A learned vocabulary of $k$ high-level concepts meant to enable communication with human stakeholders [11], and (ii) a simulatable [12] classifier whose predictions depend solely on those concepts. Formally, a CBM $f : \mathbb{R}^d \to [c]$, with $[c] := \{1, \dots, c\}$, maps instances $\mathbf{x}$ to labels $y$ by measuring how much each concept activates on the input, obtaining an activation vector $\mathbf{z}(\mathbf{x}) := (z_1(\mathbf{x}), \dots, z_k(\mathbf{x})) \in \mathbb{R}^k$, aggregating the activations into per-class scores $s_y(\mathbf{x})$ using a linear map [1], and then passing these through a softmax, i.e.,

$$s_y(\mathbf{x}) := \sum_j w_{yj} z_j(\mathbf{x}), \qquad p(y \mid \mathbf{x}) := \text{softmax}(\mathbf{s}(\mathbf{x}))_y. \tag{1}$$

Each weight $w_{yj} \in \mathbb{R}$ encodes the relevance of concept $z_j$ for class $y$. Now, it is possible to extract human understandable local explanations disclosing how different concepts contributed to any given decision $(\mathbf{x}, y)$ by looking at their activations and their associated weights, thus abstracting away the underlying computations. This yields explanations of the form $\{(w_{yj}, z_j(\mathbf{x})) : j \in [k]\}$ that can be readily summarized and visualized [13, 14]. GlanceNets inherit this feature.

Crucially, CBMs are only interpretable insofar as their concepts are and existing approaches implement special mechanisms to this effect [1, 3, 5]. In our case, we consider Concept Bottleneck models (CBNMs) [15, 4], which align the concepts using concept-level supervision, possibly obtained from a separate source, like ImageNet [16]. From a statistical perspective, this seems perfectly sensible: if the supervision is unbiased and comes in sufficient quantity, and the model has enough capacity, this strategy *appears* to guarantee the learned and ground-truth concepts to match.

Unfortunately, concept-level supervision alone is *not* sufficient to guarantee interpretability. [7] have demonstrated through simple examples that concepts acquired by CBNMs pick up spurious properties of the data. This phenomenon is known as *concept leakage*. Intuitively, leakage occurs because in CBNMs the concepts end up unintentionally capturing distributional information about unobserved aspects of the input, failing to provide well-defined semantics. However, a clear definition of leakage is missing, and so are strategies to prevent it: a key contribution of our paper is showing that leakage can be understood from the perspective of domain shift and dealt with using open-set recognition [10].

## 3    Interpretability and Leakage

The main issue with heuristics used by CBMs is that they are based on unclear notions of inter-pretability. In order to develop effective algorithms, we propose to view interpretability as a form of *alignment* between the machine's representation and that of its user.

**Interpretability.** We henceforth focus on the (rather general) generative process shown in Fig. 1 the observations $\mathbf{X} \in \mathbb{R}^d$ are caused by $n$ generative factors $\mathbf{G} \in \mathbb{R}^n$, themselves caused by a set of confounds $\mathbf{C}$ (including the label $Y$ [17]). Notice that the generative factors *can* be statistically dependent due to the confounds $\mathbf{C}$, but as noted by [18], the total causal effect [19] between $G_i$ and $G_j$ is zero for all $i \neq j$. The generative factors capture all information necessary to determine the observation [20], so the goal is to learn concepts $\mathbf{Z} \in \mathbb{R}^k$ that recover them. The variable $T$ will be introduced later on.

We posit that a (learned) representation is interpretable if it supports *symbolic communication* between the model and the user, in the sense that it shares the same (or similar enough) semantics to user's representation. The latter is however generally unobserved. Then, we make a second assumption that *some* of the generative factors $\mathbf{G}_I \subseteq \mathbf{G}$ are interpretable to the user and can be used as a proxy for the user's internal representation. Naturally, not all generative factors are interpretable [21], but in many applications some of them are, e.g., the hair color or noise size in CelebA [22].

**Interpretability as alignment.** Under this assumption, if the variables $\mathbf{Z}_J \subseteq \mathbf{Z}$ are *aligned* to the generative factors $\mathbf{G}_I$ by a map $\alpha : \mathbf{g} \mapsto \mathbf{z}_J$ that preserves semantics, they are themselves interpretable. One desirable property is that $\alpha$ does not "mix" multiple $G$'s into a single $Z$. This is however insufficient: we wish the map between $G_i$ and its associated factor $Z_j$ to be "simple", so as to *conservatively* guarantee that it preserves semantics. Motivated by this, we say that $\mathbf{Z}_J$ is *aligned* to $\mathbf{G}_I$ if: **(i)** the map $\alpha$ *disentangles* $\mathbf{Z}_J$, i.e., according to [18], each latent factor $Z_j$ varies only upon variations of a single generative factor $G_i$, and **(ii)** the scalar map $\alpha_j(g_i)$ is *monotonic*.

**Achieving alignment with concept-level supervision.** It has been shown that disentanglement cannot be achieved in the purely unsupervised setting without strong inductive bias [23]. This immediately entails that alignment is also impossible in that setting, highlighting a core limitation of CBMs with no concept supervision. However, disentanglement can be attained if supervision about the generative factors is available, even only for a small percentage of the examples [24]. Thus, following CBNMs, we seek alignment by leveraging concept-level supervision.

**Interpretability and concept leakage.** Intuitively, concept leakage occurs when a model is trained on a data set on which (*i*) some generative factors $\mathbf{G}_V \subset \mathbf{G}$ vary, while the others $\mathbf{G}_F = \mathbf{G} \setminus \mathbf{G}_V$ are fixed, and (*ii*) the two groups of factors are statistically dependent. For instance, in the even vs. odd experiment of [7], no training examples are annotated with concepts besides 4 and 5. CBNMs with access to supervision on $\mathbf{G}_V$ tend to acquire a latent representation that approximates these factors, and that because of (*ii*) correlates with the fixed factors $\mathbf{G}_F$.

In contrast with previous assessments [7, 8], we notice that point (*i*) can be viewed as a special form of domain shift: the training examples are sampled from a ground-truth distribution $p(\mathbf{X}, \mathbf{G} \mid T = 1)$ in which $\mathbf{G}_F$ is approximately fixed, e.g., $p(\mathbf{G}_F \mid T = 1) = \delta(\mathbf{g}'_F)$ for some vector $\mathbf{g}'_F$, and the test set from a different distribution $p(\mathbf{X}, \mathbf{G} \mid T = 0)$ in which $\mathbf{G}_F$ is no longer fixed. Here, $T$ is a random variable that selects between training and test distribution, see Appendix C. Since regular CBMs have no strategy to cope with domain shift, they fail to adapt when this occurs.

Motivated by this, we propose then to tackle concept leakage by designing a CBM specifically equipped with strategies for detecting instances that do not belong to the training distribution using open-set recognition [10], inferring the value of $T$. This strategy proves very effective, as shown by our evaluation (Section 5.2).

## 4    GlanceNets

GlanceNets combine a VAE-like architecture [25, 26] for learning disentangled concepts with a prior and classifier designed for open-set prediction [27]. In order to accommodate for non-interpretable factors, the latent representation of GlanceNets $\mathbf{Z}$ is split into two: (i) $k$ concepts $\mathbf{Z}_J$, aligned to the *interpretable* generative factors $\mathbf{G}_I$, that are used for prediction, and (ii) $\bar{k}$ *opaque* factors $\mathbf{Z}_{\bar{J}}$ that are only used for reconstruction. Specifically, a GlanceNet comprises an encoder $q_\phi(\mathbf{Z} \mid \mathbf{X})$ and a

decoder $p_\theta(\mathbf{X} \mid \mathbf{Z})$, both parameterized by deep neural networks, as well as a classifier $p_W(Y \mid \mathbf{Z}_J)$ feeding off the interpretable concepts only. Following other CBMs, the classifier is implemented using a dense layer with parameters $W \in \mathbb{R}^{v \times k}$ followed by a softmax activation, and the most likely label is used for prediction. The overall architecture is shown in Fig. 1.

In contrast to regular VAEs, GlanceNets associate each class to a prototype in latent space through the prior $p(\mathbf{Z} \mid \mathbf{Y})$, which is conditioned on the class and modelled as a *mixture of gaussians* with one component per class. The encoder, decoder, and prior are fit on data so as to maximize the evidence lower bound, defined as [28] $\mathbb{E}_{p_D(\mathbf{x},y)}[\mathcal{L}(\theta, \mathbf{x}, y; \beta)]$ with:

$$\mathcal{L}(\theta, \mathbf{x}, y; \beta) := \mathbb{E}_{q_\phi(\mathbf{z}|\mathbf{x})}[\log p_\theta(\mathbf{x} \mid \mathbf{z}) + \log p_W(y \mid \mathbf{z}_J)] - \beta \cdot \mathsf{KL}(q_\phi(\mathbf{z} \mid \mathbf{x}) \,\|\, p(\mathbf{z} \mid y)) \quad (2)$$

Here, $p_D(\mathbf{x}, y)$ is the empirical distribution of the training set $D = \{(\mathbf{x}_i, y_i) : i = 1, \ldots, m\}$. As mentioned in Section 3, learning disentangled representations is impossible in the unsupervised i.i.d. setting [23]. Following [24], and similarly to CBNMs, we assume access to a (possibly separate) data set $\widetilde{D} = \{(\mathbf{x}_\ell, \mathbf{g}_{I,\ell})\}$ containing supervision about the *interpretable* generative factors $\mathbf{G}_I$ and integrate it into the ELBO by replacing the per-example loss $\mathcal{L}$ in Eq. (2) with:

$$\mathcal{L}(\theta, \mathbf{x}, y; \beta) + \gamma \cdot \mathbb{E}_{p_{\widetilde{D}}(\mathbf{x},\mathbf{g})} \mathbb{E}_{q_\phi(\mathbf{z}|\mathbf{x})}[\Omega(\mathbf{z}, \mathbf{g})] \quad (3)$$

where $\gamma > 0$ controls the strength of the concept-level supervision. Following [24], we implement this term using the average cross-entropy loss $\Omega(\mathbf{z}, \mathbf{g}) := -\sum_k g_k \log \sigma(z_k) + (1 - g_k) \log(1 - \sigma(z_k))$, where the annotations $g_k$ are rescaled to lie in $[0, 1]$ and $\sigma$ is the sigmoid function.

In order to tackle concept leakage, GlanceNets integrate the open-set recognition strategy of [27]. This strategy identifies out-of-class inputs by considering the class prototype $\mu_y := \mathbb{E}_{p(\mathbf{z}|y)}[\mathbf{z}]$ in $\mathbb{R}^k$ defined by the prior distribution and the decoder $p_\theta(\mathbf{x}|\mathbf{z})$. During training, GlanceNets use the training data to estimate: (*i*) a distance threshold $\eta_y$, which defines a spherical subset in the latent space $\mathcal{B}_y = \{\mathbf{z} : \|\mu_y - \mathbf{z}\| < \eta_y\}$ centered around the prototype of class $y$, and (*ii*) a maximum threshold on the reconstruction error $\eta_{thr}$. If new data points have reconstruction error above $\eta_{thr}$ or they do not belong to any subset $\mathcal{B}_y$, they are inferred as open-set instances, i.e., $\hat{T} = 0$. This procedure is illustrated in Fig. 1.

**Remark.** Other disentanglement strategies can be naturally included in GlanceNets to increase its alignment, e.g., with various methods in [29]. Since our experiments already show substantial benefits for GlanceNets building on $\beta$-VAEs, we leave these extensions to future work.

## 5 Empirical Evaluation

### 5.1 GlanceNets achieve better alignment than CBNMs

In a first experiment, we compared GlanceNets with CBNMs on a classification tasks for which supervision on the generative factors is available. In order to evaluate the impact of this supervision on the different competitors, we varied the amount of training examples annotated with it from $1\%$ to $100\%$. For each increment, we measured prediction performance using accuracy, alignment and explicitness using the lasso variant of DCI. Additional details are reported in Appendix B.

**Data set.** We carried out our evaluation on a very challenging real-world data set. _CelebA-64_ [22] is a collection of $64 \times 64$ RGB images of over 10k celebrities. Images are annotated with 40 binary generative factors including hair color, presence of sunglasses, *etc*. Since we are interested in measuring alignment, we considered only those 10 factors that CBNMs can fit well (in the Appendix). We also dropped all those examples for which hair color is not unique, obtaining approx. $127k$ examples. We kept the original train/validation/test split, as in Ref. [22], and we generated the labels $y$ clustering over the 10 binaries attributes using the algorithm in [30], for a total of 4 class labels.

**Results and discussion.** The results of this first experiment are reported in Fig. 2. In addition to alignment, we also report explicitness [31], which measures how well the linear regressor employed by DCI fits the generative factors. The higher, the better. Details on its evaluation are included in Suppl. Material. The plots clearly show that, although the two methods achieve high and comparable accuracy in all settings, GlanceNets attain better alignment in almost all percentages of attribute supervision, with a single exception in CelebA using low values of supervision. The gap is evident with full supervision, and GlanceNets still attain overall better scores in the 25% and 50% regime.

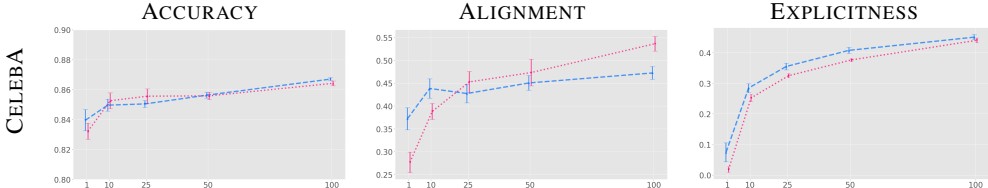

Figure 2: **GlanceNets are better aligned than CBNMs.** Each column reports a different metric. The horizontal axes indicate the % of training examples for which supervision on the generative factors is provided. We compared **GlanceNets** with **CBNMs**.

On the other hand, performance are lower, but comparable, with 10% supervision. The case at 1% refers to an extreme situation where both CBNMs and GlanceNets struggle to align with generative factors, as is clear also from the very low explicitness.

## 5.2 GlanceNets are leak-proof

Next, we evaluated robustness to concept leakage in the MNIST scenarios, for which generative factors are entangled. In the experiment, we compare GlanceNets with a CBNM and a modified GlanceNet where the OSR component has been removed (denoted CG-VAE).

**Leakage due to unobserved entangled factors.** We start by replicating the experiment of Mahinpei et al. [7]: the goal is to discriminate between even and odd MNIST images using a latent representation $\mathbf{Z} = (Z_4, Z_5)$ obtained by training (with complete supervision on the generative factors) *only* on examples of $4$'s and $5$'s. Leakage occurs if the learned representation can be used to solve the prediction task better than random on a test set where all digits except $4$ and $5$ occur. During training, we use the digit label for conditioning the prior $p(\mathbf{Z} \mid \mathbf{Y})$ of the GlanceNet. More qualitative results are collected in Appendix D.

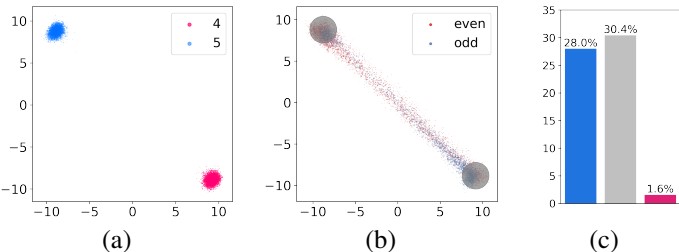

Figure 3: **GlanceNets are leak-proof on MNIST.** (*a*) Training set embedded by GlanceNet with $\beta = 100$; axes indicate $z_4$ and $z_5$ and color the concept label, i.e., $4$ vs. $5$. (*b*) Latent representations of the test images, divided in even vs. odd. Every ball in light gray denotes the region $\mathcal{B}_y$. (*c*) Information Leakage performances of the considered models: CBNM, CG-VAE and GlanceNet.

Fig. 3 (a, b) illustrates the latent representations of the training and test set output by a GlanceNet. Here the OSR kicks in: if an input is identified as open-set, $T$ is predicted as $0$ by the OSR component and the input is rejected. In all leakage experiments, we implement rejection by predicting a random label. We measure leakage by computing the difference in accuracy between the classifier and an ideal random predictor, i.e., $2 \cdot |\mathrm{acc} - \frac{1}{2}|$: the smaller, the better. The results, shown in Fig. 3 (c), show a substantial difference between GlanceNet and the other approaches. Consistently with [7], CBNMs are affected by a considerable amount of leakage. This is not the case for our GlanceNet: most (approx. $85\%$) test images are correctly identified as open-set and rejected, leading to a negligible amount of leakage. The results for CG-VAE also indicate that removing the open-set component from GlanceNets dramatically increases leakage back to more than CBNMs.

## 6 Related Work

Concepts lie at the heart of AI [32] and have recently resurfaced as a natural medium for communicating with human agents [11]. In XAI, they were first exploited by post-hoc approaches like

TCAV [33], which can however be unfaithful to the model's reasoning [34–36]. CBMs, including GlanceNets, avoid this issue by leveraging concepts for computing their predictions. Existing CBMs [2, 3, 37, 38, 1, 15, 4, 5] seek concept interpretability using heuristics, and the quality of concepts they acquire has been called into question [39, 40, 7, 8]. We show that disentangled representation learning helps in this regard.

Despite playing a large role in disentangled representation learning [41–43], interpretability has never ben explicitly linked to disentanglement. Moreover, existing approaches make no distinction between interpretable and non-interpretable generative factors and generally ignore human factors [12, 44], and specifically disentanglement does not require the map between generative and latent factors to preserve semantics. The work of Kazhdan et al. [45] is the only one that explicitly compares disentangled representations and concepts, however it makes no attempt at linking the two notions. Our work fills this gap.

## Acknowledgments and Disclosure of Funding

The research of ST and AP was partially supported by TAILOR, a project funded by EU Horizon 2020 research and innovation programme under GA No 952215.

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

# A   Implementation details

All experiments were implemented using Python 3 and Pytorch [46] and run on a server with 128 CPUs, 1TiB RAM, and 8 A100 GPUs. GlanceNets were implemented on top of the `disentanglement-pytorch` [47] library. Code for the complete experimental setup is available on GitHub, and will be released upon acceptance. For each experiment, we used exactly the same architecture and number of latent variables for both GlanceNets and CBNMs to ensure a fair comparison.

## A.1   GlanceNet and CBNM Architectures

In all experiments, we used exactly the same architecture and number of latent variables for both GlanceNets and CBNMs to ensure a fair comparison.

**Encoder architectures:**

- *CelebA*: We leveraged the architecture of Ghosh et al. [48], which is a common reference for VAE models on CelebA-64 [49]. The encoder is composed of four convolutions of depth 128, 256, 512, 1024 respectively, all with kernel size of 5, stride of 2, followed batch normalization and ReLU activation.

The models had exactly as many latent variables as generative factors for which supervision is available, which in our data set is 10.

Table 1: Structure of the encoder network used for CelebA.

| INPUT SHAPE | LAYER TYPE | PARAMETERS | FILTER | ACTIVATION |
|---|---|---|---|---|
| $(64, 64, 3)$ | Convolution | depth=128, kernel=5, stride=2 | BatchNorm | ReLU |
| $(30, 30, 128)$ | Convolution | depth=256, kernel=5, stride=2 | BatchNorm | ReLU |
| $(13, 13, 256)$ | Convolution | depth=512, kernel=5, stride=2 | BatchNorm | ReLU |
| $(5, 5, 512)$ | Convolution | depth=1028, kernel=5, stride=2 | BatchNorm | ReLU |
| $(1, 1, 1028)$ | Flatten | | | |
| $(1, 1028)$ | Linear | dim=10+10, bias = True | | |

**Decoder architecture:** All models share the same decoder architecture, obtained by stacking:

- A 2D convolution on the latent space with a filter depth of 256, kernel size of 1, and stride of 2, followed by the ReLU activation;
- Five transposed 2D convolutions of depth 256, 256, 128, 128, 64, 64, and `num_channels`, respectively, all with kernel of size 4 and stride 2.

Here, `num_channels` is 3. The shape of the last layer was chosen so as to match the dimension of the input image. Additional details can be found in the various Tables in this appendix.

Table 2: Structure of the decoder network.

| INPUT SHAPE | LAYER TYPE | PARAMETERS | ACTIVATION |
|---|---|---|---|
| $(\dim(\mathbf{z}))$ | Unsqueeze | | |
| $(\dim(\mathbf{z}), 1, 1)$ | Convolution | depth=256, kernel=1, stride=2 | ReLU |
| $(256, 1, 1)$ | Deconvolution | depth=256, kernel=4, stride=2 | ReLU |
| $(256, 2, 2)$ | Deconvolution | depth=128, kernel=4, stride=2 | ReLU |
| $(128, 6, 6)$ | Deconvolution | depth=128, kernel=4, stride=2 | ReLU |
| $(128, 14, 14)$ | Deconvolution | depth=64, kernel=4, stride=2 | ReLU |
| $(64, 30, 30)$ | Deconvolution | depth=64, kernel=4, stride=2 | ReLU |
| $(64, 62, 62)$ | Deconvolution | depth=num_channels, kernel=4, stride=2 | |

## A.2   Supervision and Training

**Concept-level supervision.** Depending on the supervision provided, only a fraction of the inputs was made available during training with their generative factors. We restricted learning to those 10 attributes that are best fit by the CBNMs, namely: `bald`, `black hair`, `brown hair`, `blonde`

`hair`, `eyeglasses`, `gray hair`, `male`, `no beard`, `smiling`, and `wearing hat`. Both CBNMs and GlanceNets are jointly trained, meaning that optimization steps for the concepts and label supervision are taken simultaneously. Whenever concept supervision is lower than 100%, for those examples without concept annotations we trained both models using label supervision only. We did not evaluate other training strategies available for CBNMs (e.g., sequential training [15]) as these appear to bring no benefit in terms of either performance nor leakage.

**Optimization setup.** In all experiments, we used the Adam optimizer [50] with default parameters $\beta_1 = 0.9$ and $\beta_2 = 0.999$. We used a batch size of 100 and annealed the learning rate from $10^{-7}$ to $\eta_{CelebA} = 10^{-4}$. To prevent overfitting, we multiplied the learning rate by a factor of 0.95 in each epoch and apply early stopping on the validation set, with a patience of 10 epochs.

Prior to training, we selected a reasonable value for the following hyper-parameters:

- $\beta$: the weight of the KL divergence in Eq. (2).

- $\gamma$: the weight of the loss on the generative factors in Eq. (3).

- $\lambda$: the weight of the cross-entropy loss over the label, which is left implicit in Eq. (2).

We achieved good performance with $\lambda = 10^3$ and $\gamma = 7 \cdot 10^3$, with the exception that we reduced the reconstruction error by 0.01. This error was restored to 1 for additional tests Fig. 7. Moreover, we cross-validated over different values of $\beta$ but we obtained better alignment performances with $\beta \approx 1$. This happens because we inject supervision on the latent factors (which is absent in regular $\beta$-VAEs [51]).

## A.3 Implementation of leakage tests

**MNIST.** For this dataset, we considered only Multi-Latyer Perceptrons instead of convolutions. Both the encoder and the decoder are composed by two linear layers with depth 128, and a dense layer connected to the latent space and to the input space, respectively. Further details are on Table 3.

For the GlanceNet we considered a latent space of dimension 10 where the supervision on the 4 and 5 digits is used to fit the $\{z_4, z_5\}$ latent factors. These two, constitute the latent subspace where leakage occurs, while the other are useful only for reconstruction. Conversely, for the CBNM we considered only two latent factors.

During training of the latent encodings, we used stochastic gradient descent with learning rate $\eta = 0.001$, reducing it by 0.95 in each epoch for both CBNMs and GlanceNets. The training was performed only on the 4 and 5 digits (in the usual training set partition for MNIST), for almost 50 epochs. Afterwards, we considered the open-set representations, restricted to $\{z_4, z_5\}$, as inputs for training a logistic regression for parity recognition. During the training, only the digits in the MNIST training set partition (exception made for 4 and 5) are considered, while performance are calculated on the test set.

Table 3: Encoder and Decoder structures for MNIST

| TYPE | INPUT SHAPE | LAYER TYPE | PARAMETERS | ACTIVATION |
|---|---|---|---|---|
| ENCODER | | | | |
| | $(28, 28)$ | Flatten | | |
| | $(784)$ | Linear | dim=128, bias=True | ReLU |
| | $(128)$ | Linear | dim=128, bias=True | ReLU |
| | $(128)$ | Linear | dim=10+10, bias=True | |
| DECODER | | | | |
| | $(\dim(\mathbf{z}))$ | Linear | dim=128, bias=True | ReLU |
| | $(128)$ | Linear | dim=128, bias=True | ReLU |
| | $(128)$ | Linear | dim=728, bias=True | |
| | $(728)$ | Unsqueeze | | |

# B  DCI framework

In our case study, we are interested into DCI maps that linearly connect the $\mathbf{z}'s$ to the $\mathbf{g}'s$. In order to evaluate alignment performances, the inverse map $\alpha^{-1} : \mathbb{R}^k \to \mathbb{R}^{n_I}$ is constructed from the latent space to the span of the $n_I$ generative factors. The latent representations and generative factors were normalized in the $[0, 1]$ interval prior to learning.

## B.1  Alignment and explicitness

The importance weights of this map are the absolute-values of the weights in the linear matrix of $\alpha^{-1}$, indicated as $B \in \mathbb{R}^{k \times n_I}$ in the main text, see Section 3. Then, the importance weights are used to evaluate the dispersion of the learned weights. To this end, we measure each Shannon entropy $H_j$ on all $k$ latent factors:

$$H_j = -\sum_{i \in 1}^{n_I} \bar{b}_{ji} \log_n \bar{b}_{ji} \quad \text{where} \quad \bar{b}_{ji} = b_{ji} \Big/ \sum_{\ell=1}^{n_I} b_{j\ell} \tag{4}$$

Then, the average alignment is calculated as:

$$\text{alignment} = 1 - \sum_{j=1}^{k} \rho_j H_j \quad \text{where} \quad \rho_j = \sum_{i=1}^{n_I} b_{ji} \Big/ \sum_{j'=1,i=1}^{k,n_I} b_{j'i} \tag{5}$$

and ranges in $[0, 1]$. Similarly, the quantity:

$$\tilde{b}_{ji} = b_{ji} \Big/ \sum_{\ell=1}^{k} b_{\ell i} \quad \text{and} \quad \tilde{H}_i = \sum_{j=1}^{k} \tilde{b}_{ji} \log_k \tilde{b}_{ji} \tag{6}$$

is the *completeness* of the latent representation, a measure akin to alignment (Eq. (5)) that quantities the degree to which each generative factor correlates with *distinct* latent factors. Alignment and completeness relate to different properties of the map: the higher the *alignment*, the more each $Z_j$ depends on variations of only a single $G_i$. On the other hand, learning multiple $Z_j$'s capturing a single $G_i$ reduces the *completeness*. As an illustrative example, consider the matrix:

$$B = \begin{pmatrix} 1 & 0 & 0 \\ 0 & 0 & 1 \\ 0 & 1 & 0 \\ 0.2 & 0 & 0 \\ 0 & 0.2 & 0 \end{pmatrix}$$

From the above definitions, one gets $alignment = 1$ and $completeness < 1$. This follows since each Shannon entropy for the alignment score is zero (as it is related to the rows), whereas the Shannon entropy for the completeness is greater than zero (it refers to the columns). Moreover, each latent variable $z_i$ depends only on the variations of a single generative factor $g_j$.

We also calculate the explicitness of the map $\alpha$, which is related to the mean squared error (MSE) of the prediction. Since the MSE for random guessing for a variable in the $[0, 1]$ interval is equal to 1/6, the explicitness becomes:

$$\text{explicitness} = 1 - 6 \cdot \text{MSE}$$

## B.2  Empirical evaluation

Since the 40 attributes in CelebA are not exhaustive for the image generation, we implemented computed DCI as follows: *(i)* we first converted the $J$ attributes $\mathbf{z}_J$ and $\mathbf{g}_J$ connected to `hair type` to a single concept $h$ and fit the model with Lasso regression to predict $g_h$ from $\mathbf{z}$. Then, *(ii)* we trained a Logistic Regression with $l1$ penalty to predict the remaining $\mathbf{g}'s$. Finally, we took both weights in *(i)* and in *(ii)* to compute the matrix $B \in \mathbb{R}^{6 \times 6}$. In this way, we determined alignment and explicitness for CelebA. We chose the lasso coefficient $\lambda = 0.01$ for both regressions.

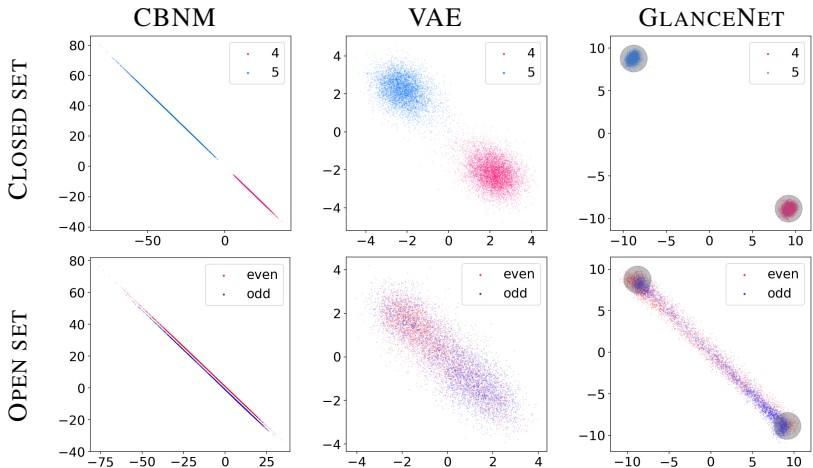

Figure 4: **Latent space representation for MNIST.** On the first row, we report the representations for 4 and 5 as fitted by CBNM, VAE and GlanceNet, respectively. On the second row, we display the scattering plot for points only belonging to the open set. For CBNM, we separated even and odd instances by $\Delta y = 2$, since their representations strongly overlap. All plots comprise only the $z_4, z_5$ axes.

## C  Open-Set Recognition Mechanism

In this section, we provide additional details on the OSR mechanism introduced in Section 3. Our method adapts the one of Sun et al. [27], which distinguishes between closed-set and open-set data points by combining a reconstruction check $\Gamma_r$ with a localization check $\Gamma_{ls}$. The overall OSR check is given by:

$$\hat{T} = \Gamma_r \wedge \Gamma_{ls} \tag{7}$$

After completing the training process, all the training instances are passed to the model to evaluate the thresholds:

- The reconstruction threshold $\eta_r$ is the maximum real number such that a fixed percentage of training examples have reconstruction error less or equal to it. At test time, given an instance $\mathbf{x}$, let $\hat{\eta} = \|\mathbf{x} - \hat{\mathbf{x}}\|^2$ be the reconstruction error. Then, $\Gamma_r = 1$ (i.e., the check passes) if the empirical reconstruction error is less than the threshold, $\hat{\eta} < \eta_r$, otherwise $\Gamma_r = 0$.

- The latent-space distance thresholds are evaluated for each class-prototype embedded in the latent space $\mu_y = \mathbb{E}_{p(\mathbf{z}|y)}[\mathbf{z}]$. For each of them, we first evaluated the relative distance between point belonging to the class $y$ and the prototype $\mu_y$. Then, we evaluated a threshold $\eta_y$ on the distances, as to include a fixed percentage of training instances into the set $\mathcal{B}_y = \{\mathbf{z} : \|\mu_y - \mathbf{z}\| < \eta_y\}$. At test time, those points that do not belong to any set $\mathcal{B}_y$ are predicted as open-set instances, i.e. $\Gamma_{ls} = 0$, otherwise $\Gamma_{ls} = 1$.

In our experiments, the threshold are obtained by fixing both reconstruction and latent space distance to keep the 95% of training data. In the case of $\eta_y$, this quota has been reached singularly for each $\mathcal{B}_y$, thus obtaining different values $\eta_y$'s from one another. Finally, combining both rejection methods we are sure the model would predict as closed-set at least the 90% of training instances.

## D  Concept Leakage in MNIST

We report here additional details for the concept leakage test on MNIST, which has been originally introduced by Margeloiu et al. [8]. The experiment has two stages:

1. At train time, the model is trained to align its representations to the concepts of 4 and 5, by passing full supervision on them. Both CBNMs and GlanceNets are allotted two latent concepts, which we denote $(Z_4, Z_5)$. There is no downstream classification task in this stage.

2. At test time, all MNIST images, excluding those of 4's and 5's, are encoded using the learned encoder and used to learn a classifier of even vs. odd digits. The performance of the resulting classifier, applied to non-$\{4, 5\}$ images, is then computed.

In this experiment, concept leakage occurs if the accuracy on the downstream task is above the $50\%$.

### D.1 Qualitative results

In Fig. 4, we show the latent space representations for different models on the MNIST leakage test, for both closed-set and open-set data points. To illustrate the contribution of our mixture prior, in addition to the CBNM and GlanceNet models, we also considered a simpler supervised VAE model. This model has the same encoder, decoder, and classifier as the GlanceNet, but uses a regular Gaussian prior[1]. We found that this model achieved a similar level of leakage to CG-VAE. We display in Fig. 5 the reconstruction of a few random examples output by GlanceNet: the reconstructions of all instances belonging to the open-set greatly deviate from the original.

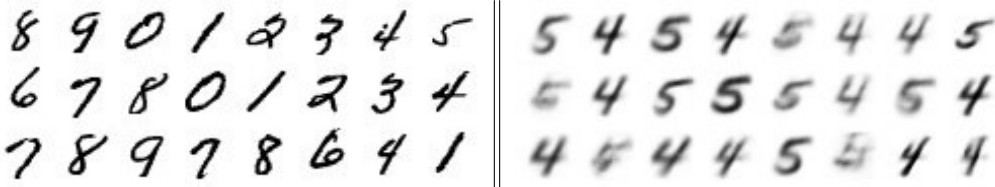

Figure 5: **MNIST reconstruction with GlanceNet.** On the left we reported the original digits, whereas on the right the reconstruction with the learned decoder. All images have been inverted in the black and white scale.

## E  Additional results for GlanceNets and CBNMs in CelebA

In this section, we discuss additional results for CBNMs vs. GlanceNets on the CelebA dataset. We first report the accuracy of the learned concepts on the supervised latent factors for both CBNMs and GlanceNets in CelebA. Then, we examine two variants of GlanceNets varying the dimension of the unsupervised factors in the latent space: a $\beta$-VAE with 20 latent factors and a $\beta$-TCVAE with 40 latent factors, [52]. This variant includes an additional loss term to Eq. (3) given by the Total Correlation (TC) of the model posterior $q_\phi(\mathbf{z}) = \mathbb{E}_{p_\mathcal{D}(\mathbf{x})}[q_\phi(\mathbf{z}|\mathbf{x})]$:

$$(\beta - 1) \cdot \mathsf{KL}\big(q_\phi(\mathbf{z})||\prod_{i=1}^{k} q_\phi(z_i)\big) \tag{8}$$

where $\beta$ denotes the strength hyper-parameter. Both the $\beta$-VAE and the $\beta$-TCVAE receive supervision only on the 10 generative factors that are fitted in the CBNM. A the end of the section, we report traversals for the models with 40 latent factors.

### E.1 Concepts Accuracy

We report the concepts accuracy for both CBNMs and GlanceNets in Fig. 6, with 10 latent dimensions and the TCVAE variant. The difference in concept accuracy between GlanceNet (both variants) and CBNMs is negligible, with GlanceNets showing slightly higher variance when the percentage of concepts supervision is very small. This highlights how, in terms of concept accuracy, the two classes of models are essentially indistinguishable, even though they are in terms of alignment.

### E.2 Performances upon variations of the latent space dimension

Similarly to the analysis in Section 5.1, we show the behavior of the metrics upon increasing the dimension of the latent space. The first variant of GlanceNets, based on $\beta$-VAE, was fitted with $\beta \approx 1$, with a latent space of dimension 20. The second variant is a TCVAE, trained with a weight of

---

[1]For the VAE model, we chose the Gaussian prior in [25], i.e., $p(\mathbf{z}) = \mathcal{N}(\mathbf{z}|0, 1)$.

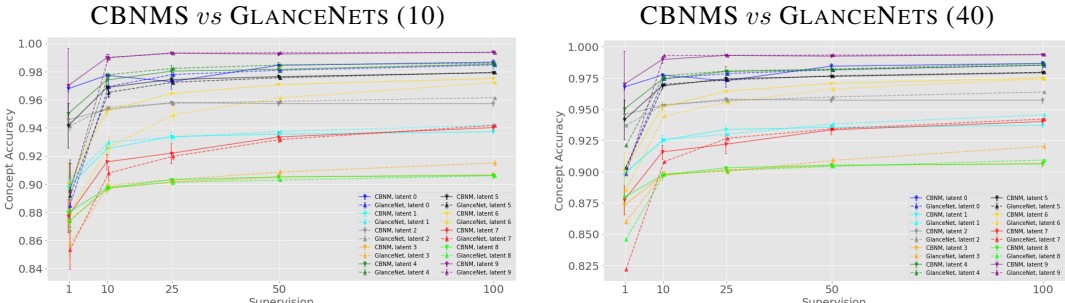

Figure 6: Concepts accuracy for CBNMs *vs* GlanceNets. Different colors refer to the distinct attributes for which supervision is provided. The solid line is reserved to CBNMs, whether GlanceNets are displayed with a dotted line. On the left, CBNMs vs GlanceNets with a latent space of dimension 10. On the right, CBNMs vs GlanceNets with TCVAE variant and a latent space of 40.

the total correlation $\beta = 10$ for all concepts supervisions, exception made for the $100\%$ run, where we found better results with $\beta = 0.5$. We measured alignment and explicitness for both variants of GlanceNets by restricting on only those 10 latent factors where supervision were provided. This is in line with the notion of alignment, since we are interested in measuring the interpretability of the model, not the disentanglment among different components.

In Fig. 7 we report the results obtained, including the original variant with 10 latent dimensions. For the $\beta$-VAE (20) and TCVAE (40) we can see the improvement provided by extending the latent space. The latter achieves particular high values of alignment.

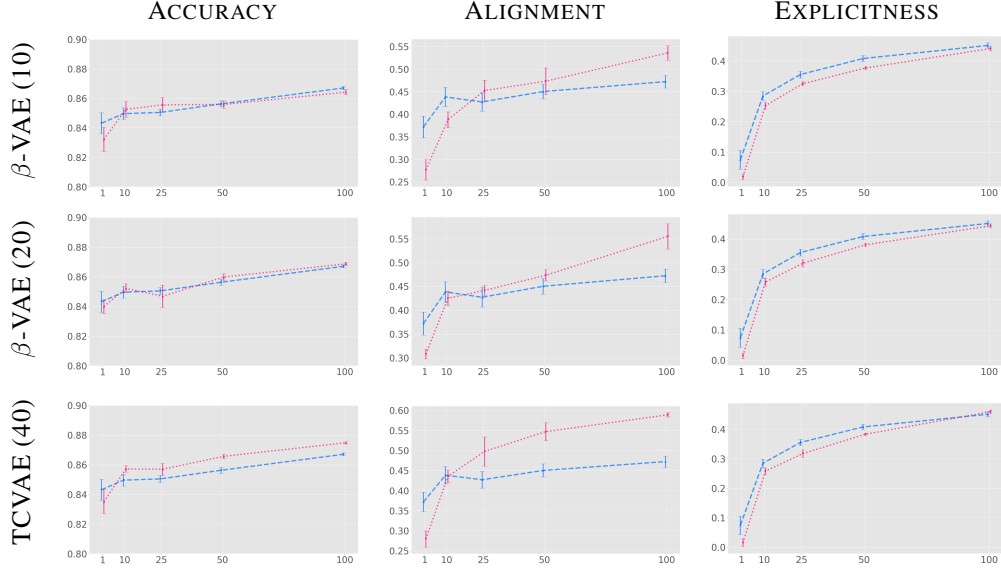

Figure 7: Accuracy, Alignment and Explicitness metrics for CBNMs *vs* GlanceNets. For each row we vary the comparison with variants of GlanceNets: $\beta$-VAE (10) refers to the model we reported in the main text, $\beta$-VAE (20) is a variant with 20 latent dimensions, and TCVAE (40) is the model based on a TCVAE with 40 latent dimensions.

## E.3 Latent traversals

We finally report in the traversals for some of the supervised attributes, obtained by the GlanceNet TCVAE with full supervision on the concepts. We excluded the traversals of the attributes HAT and BALD since the generator failed to reproduce them faithfully. The others are well captured by the model, as we reported in Fig. 8.

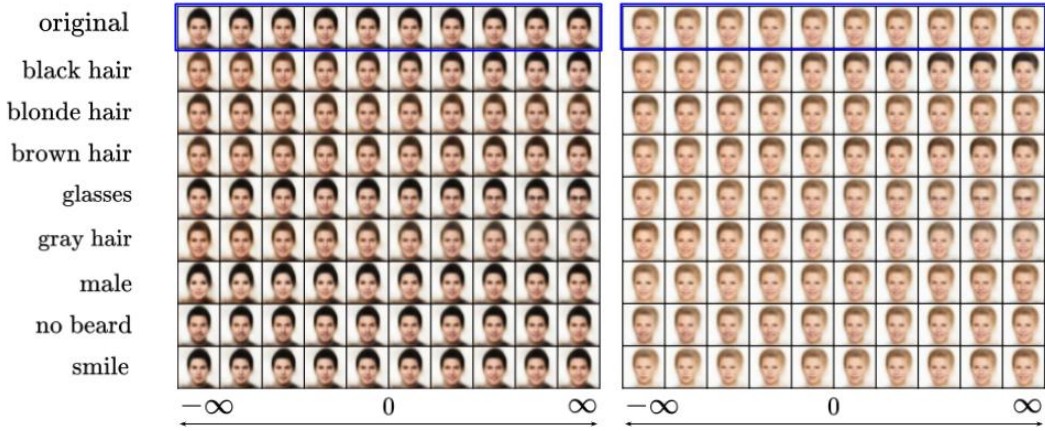

Figure 8: Latent traversals on two test images. In each row, we report the result of changing a single latent factor $Z_i$ (from $-5$ to $+5$) while keeping fixed the others.

