# OpenReview forum: "GlanceNets: Interpretable, Leak-proof Concept-based Models"
_NeurIPS.cc/2022/Workshop/nCSI — nCSI WS @ NeurIPS 2022 Oral_

### Official Review · Reviewer_HNvC · 2022-10-12
**Addresses the issues of concept-based models**

**Rating:** 2
**Confidence:** 2

**Review:**

**Summary**: This paper tackles two important issues of concept-based models: interpretability and concept leakage. Both issues arise from a lack of a widely accepted definition, resulting in an inability to concretely quantify the problem. The paper proposes to define interpretability in terms of alignment, where the goal is to learn a representation for which each variable monotonically maps to a generative factor of the data. They propose GlanceNet, a new type of concept-based model that learns the representation using a VAE architecture. Concept leakage is addressed by rejecting samples that are too far away from labels in the latent space or if the reconstruction loss is too high. GlanceNets are shown empirically to have better alignment and resistance to leakage compared to the competitor, CBNM.

**Strengths**:
1. The problem is clearly motivated, and a solution could indeed be quite useful from the perspective of explainable AI. I particularly like the approach of working with causal assumptions, which perhaps could be expanded more formally. The connection with disentanglement representation learning is also quite interesting.

2. The GlanceNet architecture and optimization appears to be an interesting novel approach to this problem. The experimental results are not overly extensive, but they are quite convincing.

**Weaknesses**:
1. The goal of setting concrete definitions for interpretability and concept leakage seems to fall short here. While Sec. 3 attempts to define these concepts for this problem setting, it is still not clear how these properties can be measured and why these definitions capture the issue at hand. There is also a lack of theoretical results, which would help rigorously ground these concepts.

2. Despite the positive experimental results, it is unclear if the approach is sound. Even under perfect training of GlanceNet, is it guaranteed that the generative factors will be recovered? Also, why does the rejection approach work for solving concept leakage? The cited paper, Mahinpei et al. (2021) seems to suggest that many approaches to avoiding concept leakages do not work, including adding latent capacity and decorrelating factors, which this paper appears to do as well. Theoretical results would help here.

3. In general, the presentation of the paper could be improved for clarity purposes. Most prominently, the work builds on several cited sources which are often leveraged without context (for example the MNIST experiment in Mahinpei et al. (2021) or the open-set recognition strategy of Locatello et al. (2020)). Providing some background on these components and how they relate to this paper could help the reader follow better. Additionally, this paper would benefit from having examples in Sec. 1 or 2 to illustrate the issues with interpretability and concept leakage, and also in Sec. 3 to explain the intuition of each of the solutions described in the paper.

In general, I think the ideas presented in this paper offer interesting insight into the issues of concept-based models and would be a great contribution to the workshop.

---

### Official Review · Reviewer_hFSd · 2022-10-15

**Rating:** 3
**Confidence:** 2

**Review:**

This paper presents a new type of CBM, called GlanceNets, that addresses two limitations of existing models: interpretability and concept leakage. GlanceNets build on existing CBMs by leveraging (1) concept-level supervision (cf. CBNMs) to achieve alignment (disentanglement?) between the generative and learned representations, thereby increasing interpretability; and (2) open-set recognition to deal with domain shift, thereby reducing concept leakage. These are implemented on top of a VAE architecture and outperform CBNMs on several datasets.

The ideas presented in this paper are very interesting and make novel steps towards addressing significant challenges with CBMs. The experiments and results are also well presented. I think this would be a great paper for the workshop but feel that some of the notions used could be described more clearly in the text to improve the work:
- How does alignment relate to disentanglement? Why/how is it better at preserving semantics?
- How is alignment measured, i.e. what are the y-axes in Figure 1?
- How does open-set recognition address the problem of concept leakage “failing to provide well-defined semantics”, i.e. where does it relate to semantics?

---

### Meta-Review · Area_Chair_FQiw · 2022-10-18

**Recommendation:** 2
**Confidence:** 3

**Metareview:**

Concept learning in the context of VAEs and interpretability are two key aspects of great relevance to the workshop. The reviewers were enthusiastic and the paper should instigate a useful discussion. I am not convinced that disentanglement alone should be sufficient to achieve alignment as broadly intended by the paper. Claims of interpretability would need to be evaluated further in more detail.

---

### Decision · Program_Chairs · 2022-10-20

Accept (Oral)